# Is Ocular Accommodation Influenced by Dynamic Ambient Illumination and Pupil Size?

**DOI:** 10.3390/ijerph191710490

**Published:** 2022-08-23

**Authors:** Hanyang Yu, Wentao Li, Ziping Chen, Mengzhen Chen, Junwen Zeng, Xijiang Lin, Feng Zhao

**Affiliations:** 1State Key Laboratory of Ophthalmology, Zhongshan Ophthalmic Center, Sun Yat-sen University, Guangzhou 510060, China; 2Huizhou Third People’s Hospital, Guangzhou Medical University, Huizhou 516000, China; 3Guangdong Wlight Visual Health Research Institute, Guangzhou 510150, China; 4Educational Equipment Center of Guangdong Province, Guangzhou 510240, China

**Keywords:** accommodation, vision therapy, dynamic illumination, myopia, pupil size

## Abstract

Purpose: We investigated ocular accommodative responses and pupil diameters under different light intensities in order to explore whether changes in light intensity aid effective accommodation function training. Methods:A total of 29 emmetropic and myopic subjects (age range: 12–18 years) viewed a target in dynamic ambient light (luminance: 5, 100, 200, 500, 1000, 2000 and 3000 lux) and static ambient light (luminance: 1000 lux) at a 40 cm distance with refractive correction. Accommodation and pupil diameter were recorded using an open-field infrared autorefractor and an ultrasound biological microscope, respectively. Results: The changes in the amplitude of accommodative response and pupil diameter under dynamic lighting were 1.01 ± 0.53 D and 2.80 ± 0.75 mm, respectively, whereas in static lighting, those values were 0.43 ± 0.24 D and 0.77 ± 0.27 mm, respectively. The amplitude of accommodation and pupil diameter change in dynamic lighting (t = 6.097, *p* < 0.001) was significantly larger than that under static lighting (t = 16.115, *p* < 0.001).The effects of light level on both accommodation and pupil diameter were significant (*p* < 0.001). Conclusion: Accommodation was positively correlated with light intensity. The difference was about 1.0 D in the range of 0–3000 lux, which may lay the foundation for accommodative training through light intervention.

## 1. Introduction

Digital eye strain (DES), also known as computer vision syndrome, comprises a range of ocular and visual symptoms. The prevalence of DES has been estimated to be 50% or more among computer users [1]. The symptoms can be subsumed under two broad categories: visual symptoms linked to accommodative or binocular vision(eyestrain, eye ache, headache and blurred vision)and external ocular symptoms linked to dry eye(burning, watery eyesand irritation) [2]. Although these symptoms are typically transient, they can cause frequent and persistent discomfort for patients. The onset of DES may be triggered by excessive, frequent and intense close-range use of digital devices [2]. Unfortunately, with the widespread use and increased variety of digital devices, the usage of digital devices among all age groups (infants and young adults) has increased significantly [3]. DES has gradually become a global public health issue [2].

Previous studies have suggested that conventional vision training, including the use of the Hart chart and flipper lenses, can alleviate the symptoms related to accommodation in DES [4]. Tosha [5] found that the discomfort of participants with high scores on the Visual Fatigue Scale was characterized by accommodative fatigue, with a higher lag of accommodation developing at a close viewing distance over time. Vasudevan [6] performed flipper training on adults with NITM (nearwork-induced transient myopia) and found that their accommodative response functions improved significantly, along with decreased NITM. Allen [7] found that flipper training can significantly improve the accommodation speed and facility of teenagers. However, these vision training techniques are boring, and patients, especially children, have poor compliance; therefore, the actual efficiency of such vision training approaches maybe insufficient. Currently, research is being carried out to develop accommodation vision therapies with higher compliance and improved efficiency.

When fixation switches from a distant object to a near object, the near-reflex response, including ocular accommodation, convergence and pupil contraction, is elicited to achieve and maintain clear single-binocular vision [8]. When changing our focus from a near target to a distant target, the reverse response occurs, including ocular disaccommodation, divergence and pupil mydriasis. This shows that there is a link between the three components of the near triad. Compared with convergence, the latency and speed of accommodation and pupil constriction are synchronized [9]. Many studies have confirmed that, under different light conditions, increased accommodative stimulation enhances accommodative responses and reduces pupil diameters [10,11,12,13,14]. It is well known that pupil constriction can be induced by light, and the pupil light reflex has been suggested to maintain optimal vision under different light conditions [15]. In addition, during the near-reflex response, convergence latency is shorter than accommodative latency, which in turn is shorter than pupil constriction latency [9]. Thus, pupil responses occur before accommodation, and accommodation occurs before convergence. So, would the accommodative response change if the pupil were stimulated by ambient illumination? In other words, would ambient illumination affect the accommodative response while inducing pupil constriction under conditions of fixed accommodative stimulation and convergence? The exploration of the above issue is helpful in improvingour understanding of the mechanism of the near-reflex response. Additionally, such exploration may lay the foundation for developing accommodation treatments which utilize changing lighting conditions. This study aimed to investigate whether light intensity affects accommodative response during the near-reflex response. Therefore, we designed a kind of artificial lighting which can be dynamically controlled by a microcomputer. The light intensity, color temperature, and light angle can dynamically, automatically and smoothly change in the range of 0~3000 lux. We recruited healthy teenagers as participants in the study to evaluate the effect of dynamic ambient illumination on pupil size and accommodation.

## 2. Methods

### 2.1. Subjects

This cross-sectional study was conducted at Zhongshan Ophthalmic Center (ZOC) in Guangzhou, China. A total of 29 healthy subjects aged between 12 and 18 years were enrolled in this study, and all the results were based on data from the participants’ right eyes. The inclusion criteria were as follows: (1) a spherical equivalent(SE) in the range of [+0.5–5.0] D and astigmatism with a value ±0.75 D; (2) a best-corrected visual acuity(BCVA) value of ≥1.0; (3) a normal range of accommodative amplitude(AA) (measured as follows: AA ≥ 15–0.25 × age (minimum AA)); (4) no accommodation abnormalities or systemic diseases—this criterion included a check of whether the participants were receiving treatment for chronic diseases that might affect accommodation; and (5) no history of amblyopia, strabismus, anisometropia or binocular abnormalities. The exclusion criteria were as follows: (1) ocular conditions that might influence the study—allergies, conjunctivitis, dry eye, etc.—and (2) having undergone refractive surgery. The study protocol was explained to all participants in accordance with the principles of the Declaration of Helsinki, and informed consent was obtained. The study protocol was approved by the Ethics Committee of Zhongshan Ophthalmic Center, Sun Yat-sen University.

### 2.2. Instruments

We reconstructed an examination room (Figure 1) with the size of 6 m ×3 m ×3.2 m (length × width × height). The walls were white, and there were no windows(to avoid outdoor light). Two lighting systems were installed in the ceiling. One lighting system was the dynamic lighting system which was independently developed and designed by our team. It was composed of 4 large, flat LED lamps. Each flat LED lamp was composed of 16 small LED lamps and had 108 independent luminous points, which can be independently controlled by the chip. The light intensity of this dynamic lighting system was 0~3000 lux. It can change smoothly and periodically, and the period can be set automatically. The other lighting system was static lighting composed of LED lamps ordered from Xia Ye Company, which can produce light of 1000 lux. These two lighting systems can be switched on and off independently. A windowed computer optometer (SRW-5001K) and anultrasound biological microscope (UBM) were arranged in the examination room to measure the accommodative response and pupil diameter, respectively. Previous studies have confirmed the validity of the instrument for measuring accommodation [16,17]. The power refractor was calibrated prior to the study.

The change in pupil size during visual processing can be influenced by many factors, including retinal illumination, an accommodative stimulus of a visual target, convergence, attention and initial pupil size, etc. Therefore, we controlled all the variables mentioned above and only observed the changes in pupil diameters and accommodation under different lighting conditions.

### 2.3. Measurements

The measurements were conducted during the same time window (9.00–12.00 a.m.) on two consecutive days.Accommodative response and pupil diameter were measured by an open-field infrared autorefractor (SRW-5001K), which allowed the targets to be viewed at any distance. In this study, the subjects’ accommodative response was measured when viewing objects at a 40 cm distance with clear vision. Firstly, their accommodative response and pupil diameter were measured seven times under static lighting of 1000 lux. After 20 min of adaptation to the dark, accommodative response was measuredunder different static lighting intensities of 5, 100, 200, 500, 1000, 2000 and 3000 lux another seven times. A duration of 2–3 min was scheduled for the subjects to adapt the illumination before the measurement.

Then, an image of the pupil and ciliary muscles was captured under the same conditions by the UBM (Model SW-3200L). Since pupillary reactions to light are synchronous, the left eye was stimulated by different lighting while an anterior segment image of the right eye was captured. Considering that dynamic illumination may result in dynamic changes to retinal images, in this study, the measurements were conducted under specific light intensities in the range of 0–3000 lx at 5, 100, 200, 500, 1000, 2000 and 3000 lx.

Slit-lamp examination, which mainly included the examination of the cornea (corneal fluoresce in staining), lens, tear film (BUT) and conjunctiva, was conducted before and after the dynamic ambient illumination. 

In this study, all the subjects were wearing fully corrected soft contact lenses to achieve the best-corrected visual acuity during the measurement. The amplitude of accommodative response was defined as the difference between the maximum and minimum values of the seven measurements. The magnitude of the pupil diameter change was defined as the same.

### 2.4. Data Analyses

This was a cross-sectional and self-controlled study. The results were presented as mean ± standard deviation. A paired *t*-test was used to analyze the difference in the amplitude of the accommodative response and pupil diameter between the two groups. One-way ANOVA was used to compare the changes in the accommodative response and the pupil diameter at different time points in the groups with different lighting treatments. In this study, the results were based on data from the participants’ right eyes, as mentioned before. The data analyses were performed using SPSS 22.0. Statistical significance was set at *p* < 0.05.

## 3. Results

A total of 29 subjects were recruited in this research, including 14 males (46.7%) and 15 females(53.3%), with an average age of 14.0 ± 4.04 years. The refractive error of the right eye was −2.61 ± 1.41 D and the amplitude of accommodation was 11.52 ± 2.28 D.

The results of accommodative response and pupil diameter changes under dynamic and static lighting are shown in Table 1. The range of the accommodative response change in the dynamic lighting group was 1.01 ± 0.53 D, and it was 0.43 ± 0.24 D in the static lighting group. The amplitude of the accommodative response change in the dynamic lighting group was significantly larger than that in the static lighting group (t = 6.097, *p* < 0.001).The range of pupil diameter change in the dynamic lighting group was 2.80 ± 0.75 mm, and it was 0.77 ± 0.27 mm in the static lighting group. Under the dynamic lighting, the change in the amplitude of pupil diameter was larger, and the difference was statistically significant (t = 16.115, *p* < 0.001).

The data of the seven repeated measurements were analyzed by one-way ANOVA, and the results showed that the pupil diameter changed significantly under different light intensities in the dynamic lighting group(*p* < 0.001),while there was no obvious change in the static lighting group with fixed light intensity. The interaction between the light mode and the pupil diameter was statistically significant (*p* < 0.001),but there was no significant difference in the total mean pupil diameter between the two groups (*p* = 0.983) (Table 2, Figure 2).

The results from the seven repeated measurements also showed that the amplitude of accommodative response change was larger in the dynamic lighting group under different light intensities, and this amplitude was statistically significant (*p* < 0.001). However, the change was small in the static lighting group with fixed light intensity, and the change was not statistically significant. In this study, the interaction between the light mode and the accommodative response was statistically significant (*p* < 0.001) (Table 2, Figure 3). Figure 4 showeds UBM image samples of the anterior segment of a subject’s eye which was captured under different light intensities while the stimulus of accommodation and convergence remained the same.

No ocular surface damage was found in the eyes of any of the participants, and BUT did not change significantly after the experiment compared with the initial slit-lamp examination. The ophthalmologist interviewed the participants after the dynamic ambient illumination to determine their subjective perceptions regarding three parameters: clarity, comfort and glare. The interviews found that the participants who underwent dynamic ambient illumination experienced either significant symptoms nor discomfort.

## 4. Discussion

This study investigated the change in human accommodative response under different ambient illumination conditions. The results showed that the amplitude of the accommodative responses induced by different light intensities of dynamic lighting was higher than that in static lighting conditions (1.01 D vs. 0.43 D).The same conclusion was drawn for the range of pupil diameter change, which was 2.8 mm in the dynamic lighting group and 0.77 mm in the static lighting group. We also found that higher light intensities were positively correlated with higher amplitudes of accommodative responses and smaller pupil diameters.

We also found that dynamic lighting with a light level between 5 and 3000 lx can induce an amplitude of accommodation of about 1.01 D. Lara [12] measured the amplitude of accommodation among subjects under two different ambient lighting conditions, with accommodative stimulation changing from −0.5 D to 9.5 D and found that the amplitude of accommodation (AA) was about 1.0 D larger under high illuminance in comparison with that under low illuminance. It was indicated that the measurement of accommodative amplitude should be performed under stable ambient lighting and that brighter illuminance can improve accommodation. In addition, a study about presbyopia at night reported that the amplitude of accommodation decreased under dark ambient lighting conditions [18], a conclusion which was similar to our findings. The amplitude of accommodation reflects the maximum accommodation, while an accommodative response reflects the accuracy and the actual usage of accommodation. Before the age of 40 years, the progression of presbyopia mainly manifests in a decrease in the amplitude of accommodation, while after the age of 40 years, this progression is often accompanied by an increase in accommodative lag, leading to aggravated symptoms [19]. In contrast to the study of Lara et al., we attempted to determine whether the accommodative response can be enhanced with an increase in light intensity and weakened with a reduction in light intensity, to achieve the goal of accommodative function training with the use of dynamic lighting. In their study, Lara et al. achieved an intense change in illumination from 1 to 30 CD/m2 by switching the lights on and off. We designed an artificial lighting system with a maximum illumination of 3000 lux, while the indoor incandescent was generally between 200 and 500 lux. The difference in the accommodative response measuredunder maximum and minimum illumination, with the same visual stimulus, represents the range which can potentially be used for accommodative training.

We performed a routine slit-lamp examination of the anterior segment at the end of the dynamic lighting exposure, and no obvious damage of the ocular surface or tear film was found, indicating that it is safe to use dynamic lighting for a certain period of time. However, the safety of using it for a longer duration remains to be tested. Moreover, Allen [7] conducted an accommodative flipper training study with teenagers for 15 min a day on three consecutive days and found that their accommodation was improved significantly. This indicates that 1 h accommodation training with dynamic lighting is enough to safely achieve the intended effect of the training. We know that high illumination may be harmful to ocular fundus and that low illumination can cause vision fatigue and unclear vision; however, under dynamic lighting, the illuminance changed all the time, with only short periods of extreme intensity. This increases the safety of the use of dynamic lighting and reduces the risk of potential damage occurring to the ocular surface and fundus.

Asthenopia (including digital eye strain and computer vision syndrome) has been very common in recent years. It may be related to the frequent and excessive use of electronic devices at close range. It mainly manifests as eye pain, headache, dry eyes, blurred vision and other uncomfortable symptoms [2]. Accommodativeinsufficiency, accommodative excess and accommodative infacility are the main causes of this condition [2]. Many patients experience alleviated symptoms and improved vision function following accommodative training [4,6,7]. However, the vision training techniques commonly used in clinics are boring, and some patients show very poor compliance. Our study revealed that dynamic lighting can alter the accommodation of teenagers, which may play a potential role in accommodation vision training approaches, in addition to approaches using flipper lenses. This study, using dynamic lighting, did not require patients to show initiative, and the training can be completed without active participation; therefore, patient compliance might increase. This may be a convenient vision training therapy for adults and teenagers who had to take online meetings or courses due to COVID-19 especially, and it may relieve their fatigue syndromes caused by long-term near work. However, the effect of dynamic lighting training and the suitable parameters for its implementation require further research.

There are some limitations in our research: Firstly, the data for pupil diameters under different lighting conditions were obtained by UBM without refractive correction. However, the accommodation was measured from eyes in a fully corrected state. Although the conditions were slightly different, Orr [13] reported that the decrease in pupil diameter with higher light levels was the same in both refractive-corrected and refractive-uncorrected subjects. However, the speed of the decrease was different. Therefore, our data should not be directly used for the analysis of accommodative miosis [20]. Secondly, we did not use the method of continuous high-frequency measurement [10]. Under stable accommodative stimulus, it was reported that the incubation period and speed of pupil contraction and dilation were different. Therefore, it remains to be further studied whether the pupil response and accommodation had corresponding changes under different illumination conditions.

In conclusion, we found that the accommodative response was different under different lighting conditions with a fixed accommodative and convergent stimulus, and the difference was about 1.0 D in the range of 0–3000 lux. These findings may lay a foundation for accommodative training through light intervention. The movement of the pupil may be controlled by the biomechanics of the iris [21], and the power of the accommodative response has been proven to come from the crystalline lens and ciliary muscle [22]. There is a certain correlation between the two; however, the specific mechanism remains to be clarified, and more mechanisms and their efficacies need to be researched further.

## Figures and Tables

**Figure 1 ijerph-19-10490-f001:**
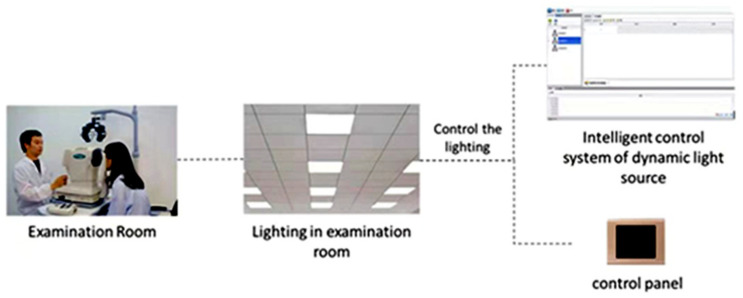
Examination room and intelligent control system of dynamic lighting.

**Figure 2 ijerph-19-10490-f002:**
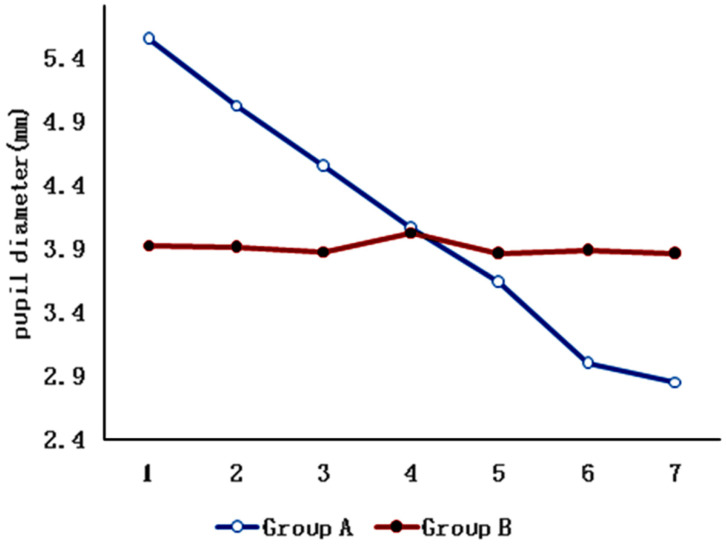
Effects of dynamic lighting (Group A) and static lighting (Group B) on pupil diameter. The abscissa from 1 to 7 comprises 5, 100, 200, 500, 1000, 2000 and 3000 lux illuminances for Group A. The abscissa from 1 to 7 comprises 7 sessions of 1000 lux illuminances for Group B.

**Figure 3 ijerph-19-10490-f003:**
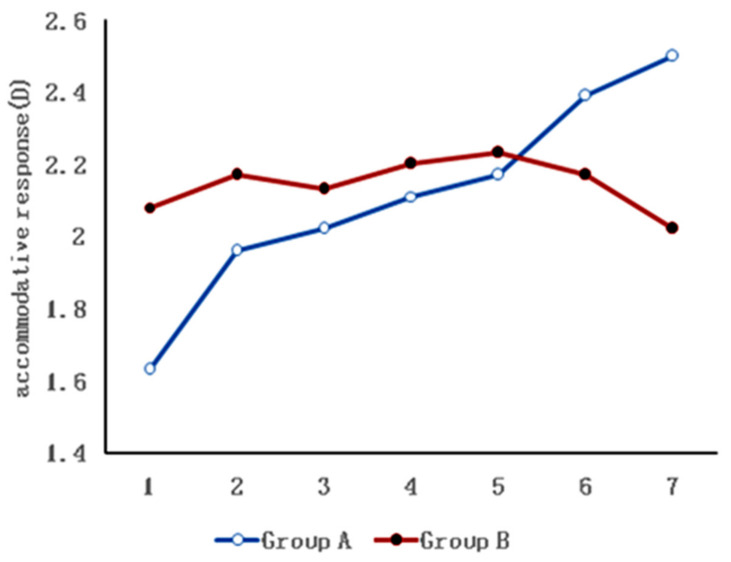
Effects of dynamic lighting (Group A) and static lighting (Group B) on accommodative response. The abscissa from 1 to 7 comprises 5, 100, 200, 500, 1000, 2000 and 3000 lux illuminances for Group A. The abscissa from 1 to 7 comprises 7 sessions of 1000 lux illuminances for Group B.

**Figure 4 ijerph-19-10490-f004:**
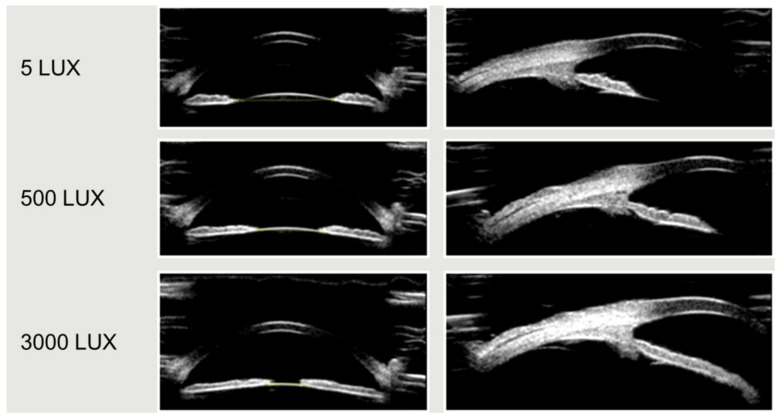
UBM image samples of the anterior segment of a subject’s eye under illuminations of 5500 and 3000 lux, respectively. The pupil shrinks significantly as the light increases.

**Table 1 ijerph-19-10490-t001:** Comparison of the effect of dynamic lighting and static lighting on human eye accommodation response and pupil diameter.

	Dynamic Lighting		Static Lighting		t Value	*p*-Value
	Mean	Standard Error	Mean	Standard Error		
Range of PD(mm)	2.80	0.75	0.77	0.27	16.115	<0.001
Range of AS(D)	1.01	0.53	0.43	0.24	6.097	<0.001

AS: accommodative response; PD: pupil diameter.

**Table 2 ijerph-19-10490-t002:** Effects of dynamic lighting and static lighting on accommodative response and pupil diameter.

No.	Dynamic Lighting			Static Lighting		
	Illuminance(lx)	AS (D)	PD (mm)	Illuminance(lx)	AS (D)	PD (mm)
1	5	1.63 ± 0.93	5.54 ± 0.76	1000	2.08 ± 0.63	3.92 ± 0.99
2	100	1.96 ± 0.87	5.02 ± 0.94	1000	2.17 ± 0.59	3.90 ± 0.98
3	200	2.02 ± 0.88	4.54 ± 0.98	1000	2.13 ± 0.59	3.87 ± 0.98
4	500	2.11 ± 0.85	4.05 ± 1.00	1000	2.20 ± 0.58	4.02 ±1.02
5	1000	2.17 ± 0.88	3.63 ± 0.96	1000	2.23 ± 0.63	3.85 ± 0.99
6	2000	2.39 ± 0.86	2.99 ± 0.67	1000	2.17 ± 0.70	3.88 ± 1.03
7	3000	2.50 ± 0.88	2.85 ± 0.60	1000	2.02 ± 0.95	3.85 ± 0.98

AS: accommodative response; PD: pupil diameter.

## Data Availability

The data used to support the findings of this study are available from the corresponding author upon request.

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
