# Peer review of "Is Ocular Accommodation Influenced by Dynamic Ambient Illumination and Pupil Size?"

_ijerph, 2022, doi:10.3390/ijerph191710490_

Round 1

Reviewer 1 Report

The authors presented an interesting study about ocular accomodation changes associated with pupil responses during a near vision task. I recommend the authors improve the readability of their paper an put more epmhasis on the study results rather than their association with other topics like COVID-19, Digital Eye Strain and/or vision therapy.

Title: Please indicate the study’s design with a commonly used term in the title or the abstract. The sentence "A vision therapy for accommodation at 3 home during COVID-19" could be removed

1. Introduction:

line 32: The study cited as #1 did not estimate DES prevalence. In my opinion, it would be better to cite a systematic review about the asthenopia prevalence. Anyway, I think that the paper's readability will improve if the authors simplify this part of the introduction and focus on the relationship between asthenopia and amplitude of accomodation.

lines 40-50: This part of the introduction may not be necessary and should be removed from the test

lines 50-60: In my opinion, it would be better to move this part of the introduction after the explanation of the near vision triad (maybe after "And may lay a foundation for the interven-80 tion of accommodation by changing lighting condition.")

lines 55-56: Please define NITM at first mention since it has not been previously defined

lines 82-83: This study did not look at whether accommodation changes could be beneficial to vision training

line 86: Healthy "adults" were not recruited, as they were from 12 yo 18 years old

In my opinion, the introduction should briefly summarize the previous research on this topic and state what the authors are expecting to contribute to this field

Finally, the authors should state specific objectives, including any prespecified hypotheses

2. Methods

This section should begin by outlining the key elements of the study desing.

Please,include recruitment, exposure, follow-up, and data collection periods, including the setting, location and dates

Authors should explain the sources and selection methods they used

Explain how the study size was determined

lines 91-92: When did you perform the cycloplegic refraction? 

lines 133-134: Why did you wait 2-3 minutes? Please explain

lines 146-153: Did you check whether all the variables followed a normal distribution?

3. Results

Including a table with a summary of the sample's clinical and demographic characteristics is strongly recommended

Figure 4: The readerx would benefit from an explanation of the differences between the photos taken with different illuminations

Did the authors check diffenrences by gender?

4. Discussion

There is a need to examine the role the accommodative lag plays in the relationship between pupil size and accommodation amplitude

lines 227-229: The slit lamp examination should be mentioned in the methods section. A more complete description of the previous examination should be included in the methods section (cycloplegic refraction, slit lamp exam, etc.).

lines 242-243: The sentence "Accommodative insufficiency, accommodative ex-242 cess and accommodative infacility were the main cause" needs a reference

lines 243-244: The sentence "Many patients alleviated the 243 symptoms and improved their vision function by accommodative training" needs a reference

lines 246-247: "Our study revealed that the dynamic lighting can improve the subject’s accommodative function, which might be a new vision therapy for accommodation" In my opinion, this statement is not supported by the results of the study

lines 253-267: In my opinion, this part of the discussion could be omitted

line 257: this hypothesis has not been previously cited

Author Response

Thank the reviewers for these precious comments concerning our manuscript entitled “Is accommodation influenced by dynamic ambient illumination and pupil size? A vision therapy for accommodation at home during COVID-19”. These comments are all valuable and very helpful for revising and improving our paper, as well as the important guiding significance to our research. We have studied comments carefully and have made corrections which we hope to meet with approval.

Title: Please indicate the study’s design with a commonly used term in the title or the abstract. The sentence "A vision therapy for accommodation at home during COVID-19" could be removed

Wearegratefulforthesuggestion.The previous title was less rigorousandinaccuracytosomeextent, since vision training has not been tested using method inthisstudy, so the title of the manuscript seems to be too far-reaching.

Wewouldliketoalterourtitleto“Accommodation can be influenced by dynamic ambient illumination and pupil size in children” with your permission”.

  1. Introduction:

line 32: The study cited as #1 did not estimate DES prevalence. In my opinion, it would be better to cite a systematic review about the asthenopia prevalence. Anyway, I think that the paper's readability will improve if the authors simplify this part of the introduction and focus on the relationship between asthenopia and amplitude of accommodation.

We thank the reviewer for this comment.Wearesorryabouttheunclearstatement. We would liketochangethesentence as following sentence.

“The prevalence of DES has be estimated to be 50% or more among computer users.”

And as showninthe first paragraph, wediscussthe symptoms can be subsumed under two broad categories: symptoms linked to accommodative visionand external ocular symptoms linked to dry eye. Weaimtoinvestigate whether the pupil diameter altered by dynamiclight would affect the accommodation response, andthatmaybehelpfulfortheaccommodationfunctionof DES.

lines 40-50: This part of the introduction may not be necessary and should be removed from the test

We thank the reviewer for this comment. Inpresentstudy, wedidnotinvestigatetheeffect ofvisiontrainingofthedynamiclight. Itisinappropriatetocitetheincreasing heavy digital near work during COVID19. Itisunnecessaryandmayreducethe paper's readability asyousay. Wewouldliketoremovethispartfromthetext.

lines 50-60: In my opinion, it would be better to move this part of the introduction after the explanation of the near vision triad (maybe after "And may lay a foundation for the interven-80 tion of accommodation by changing lighting condition.")

Wearegratefulforthesuggestionandhaveconsidereditcarefully. After removing the second paragraph(line40-50), weputforwardthetopicof DES inthefirst paragraph anddiscusstheeffectandlimitationof conventional vision training inline 51-61. Weraisethe explanation of the near vision triad behindthetwoparagraphs.

Ofnote, only the traditional visual function training methods are using in clinicalpractices. The training methods actdirectlyontheaccommodationratherthanutilizing themechanismof near triad. Based on the principle of near-reflex response, we wanted to see whether the pupil diameter altered by dynamiclight would affect the accommodation response. So weguesstheorderinpresentstudymaybesuitable.

lines 55-56: Please define NITM at first mention since it has not been previously defined

wearesorryaboutthemistake. Thedefinitionof NITM (Nearwork-Induced Transient Myopia) hasbeenput in the first mention place in line56

lines 82-83: This study did not look at whether accommodation changes could be beneficial to vision training

We thank the reviewer for this comment. As you say,thispresentationisless rigorousandinaccuracy. Wewouldliketodeletethissentenceasyoursuggestion.

line 86: Healthy "adults" were not recruited, as they were from 12 yo 18 years old

weareverysorryaboutthemistake. wewouldliketochangethiswordto “teenagers”

In my opinion, the introduction should briefly summarize the previous research on this topic and state what the authors are expecting to contribute to this field

Finally, the authors should state specific objectives, including any prespecified hypotheses

We thank the reviewer for this comment. The original presentation isnotclear enough on the topic. We involve COVID-19 whichleadtoconfusiontothereaderstoknowthemaintopicofthisarticle. Sowedeletethispartofcontentasyoursuggestioninordertooutlinethemaintopicand emphasize our prespecified hypotheses that pupil diameter altered by dynamiclight would affect the accommodation response, andthatmaybehelpfulfortheaccommodationfunctionof DES. After removing the secondparagraph(line40-50), wewrote a review of DES inthefirst paragraph anddiscusstheeffectandlimitationof conventional vision training inthesecond paragraph. Weraisethe explanation of the near vision triad behindthetwoparagraphs. Ofnote, only the traditional visual function training methods are using in clinicalpractices. The training methods actdirectlyontheaccommodation.Based on the principle of near vision triad, we wanted to see whether the pupil diameter altered by dynamiclight would affect the accommodation response.

Above is the briefly summarize of the previous research on this topic. And we state specific objectives, including prespecified hypotheses in the last paragraph inline76-83, thatisthemaincontributionweexpecttoaccomplishinpresentstudy.

We are alwaysreadytoreviseourarticle, if you consider there is anywhere that needs to be modified

line76-83:So, would accommodative response changes when the pupil is stimulated by ambient illumination? In other words, would ambient illumination affect accommodative response while it induces pupil constriction under conditions of fixed accommodative stimulation and the convergence? The exploration of the above issue is helpful to improve the understanding of the mechanism of the near-reflex response. And may lay a foundation for the intervention of accommodation by changing lighting condition. This study was aimed to investigate whetherlight intensity may affect accommodative response during near reflex

  1. Methods

This section should begin by outlining the key elements of the study desing.

Please,include recruitment, exposure, follow-up, and data collection periods, including the setting, location and dates

 We thanktoreviewforthisgoodcomment. We may not have made it clear. Thepresentstudy is a cross-sectional study. So after obtaining approval from the ZOC ethical committee prior to the initiation of the study, werecruittheappropriateparticipatebyposter. We would like to outline the key elements of the study byaddingthissentenceinline92-93.

This cross-sectional study was conducted at Zhongshan Ophthalmic Center (ZOC) in Guangzhou, China

Authors should explain the sources and selection methods they used

Weare very gratefulforthis good comment. Weusethe SRW-5001K and an ultrasound biological microscope (UBM) toexamine the accommodative response and pupil diameter. Infact, SRW-5001K can measure accommodative response and pupil sizeinthesametime. Continuouson-line analysis of the ring provides high (up to 60 Hz) temporal resolution of the refractive state toan accuracy of ,0.001 D. Pupil size can also be analysed to a resolution of ,0.001 mm.

We want to use UBM to visualize the alteredimage of pupil and ciliary muscles under different illumination, so the previous expression may be inaccurate. The measurement of pupil diameter as same as the accommodation response were through SRW-5001K. UBM shows the pupil and accommodation status in different illumination. The pupil data measured by UBM is almost the same as that of SRW-5001K, so it is enough to keep one of them. Wewouldliketorevisethesentenceinline141 andline134/137/145 asshownbelow:

line141: Then the pupil and ciliary muscles diameter was measuredimage was captured under the same conditions by UBM (Model SW-3200L).

line134:Accommodative response and pupil diameter was measured by….

line137:their accommodative response and pupil diameter were measured seven times….

line145: the pupil diameter anterior segment image of the right eye was measured.

Andwewanttoaddsentenceasfollowinline120-122.

Previous studies have confirmed the validity of the instrument measuring accommodation[1, 2]. The power refractor was calibrated priorto the study.

Explain how the study size was determined

Sample size estimations were based on the equation for sample size of independent t-test for two groups. In terms of accommodative response, we assumed the significance level α=0.05, power (1-β)=0.8, the standard deviation for dynamic lighting group (σ1) and static lighting group (σ2) is 0.4 and 0.3, respectively, and the mean difference (δ) is 0.38. 14 subjects per group was estimated. In terms of pupil diameter, we assumed α=0.05, (1-β)=0.8, σ1=0.5, σ2=0.3, and δ=0.5, and obtained an estimation of 11 subjects per groups. Therefore, the larger one (14 subjects per group) was selected. A loss of follow-up rate 5% was considered (althoughthepresentstudyisacross-sectional study, wewanttotakealong-term follow-up, sowechoosethissamplesize). Finally, the sample size of this study was estimated as 30 subjects.And 1 participateis unable to cooperate to complete the wholeexamine.

lines 91-92: When did you perform the cycloplegic refraction? 

 We thanktoreviewforthisgoodquestion. We preformthe cycloplegic refraction beforetheexperimentnearlya week ahead inordertogetthe fully corrected soft contact lenses to achievethe best corrected visual acuity during the measurement. We would liketodeletethissentenceinordertoimprovethe readability ofthearticle.

lines 133-134: Why did you wait 2-3 minutes? Please explain

Inpreviousstudy[3], whentheauthorinvestigatethe accommodation microfluctuations in emmetropesand myopes. Targets were presented to the subjects in a random order with a break of 3 minutes between conditions so that retinal adaptation effects were minimized. Inpresentstudy, 2-3min isenoughfortheparticipatetoadaptthenovelillumination.

lines 146-153: Did you check whether all the variables followed a normal distribution?

 Yes,all the variables followed a normal distribution.

  1. Results

Including a table with a summary of the sample's clinical and demographic characteristics is strongly recommended

We thanktoreviewforthisgoodcomment, the presentstudyisa cross-sectional study using self-controlled. All the participate recruit inthisstudy in accordance withtheinclusion criteriaasshownin line95-106:

inclusion criteria was described as below: 1) The spherical equivalent(SE) was in a range of [+0.5 to 5.0] D and the astigmatism was between ±0.75D;2)The best corrected visual acuity(BCVA) was≥ 1.0;3)A normal range of accommodative amplitude(AA) ( AA≥ 15-0.25× age (minimum AA)) was measured;…..

And we summarize the sample's clinical and demographic characteristics intheresultsin line167-169:

A total of 29 subjects were recruited in this research, including 14 males (46.7%) and 15 females (53.3%) with an average age of 14.0±4.04 years. The refractive error of the right eye was -2.61±1.41D and the amplitude of accommodation was 11.52±2.28D.

Figure 4: The readers would benefit from an explanation of the differences between the photos taken with different illuminations

Weare very gratefulforthis good comment. We enrich theexplanationof the differences between the photos taken with different illuminations below Fig4 as your suggestion.

Did the authors check diffenrences by gender?

Yes,we havecheck the gender differences. There was no not statistically significant betweengender.

  1. Discussion

There is a need to examine the role the accommodative lag plays in the relationship between pupil size and accommodation amplitude

Wearegratefulforthiscomment. Thepresentstudyaimstoinvestigatewhetherthe dynamic lighting can alter the accommodationofteenagers, sowerecruitparticipatewithnormalaccommodationasshowninmethods. Theassociationbetween accommodative lag andeffectoftheaccommodationresponsetrainingassameasthe variational pupilsizeisinteresting. Andaswediscussinline221-223, Before 40, the progression of presbyopia was mainly manifested in the decrease of amplitude of accommodation, while after 40, this progression was often accompanied by the increase accommodative lag. Theactualeffectofthe dynamic lighting indifferentcrowdwithdifferentaccommodationfunctionneedtobefurtherresearch. The corresponding work would conductinthenearfutureandwillpublishitatalatertime. Thepresentstudyisthe piror study of follow-up research.

lines 227-229: The slit lamp examination should be mentioned in the methods section. A more complete description of the previous examination should be included in the methods section (cycloplegic refraction, slit lamp exam, etc.).

cycloplegic refraction: Instill 2 drops of 1% cylopentoloate once every 5-10 minutes, and check whether the ciliary muscle has been paralyzed at nearly 30 minutes. Directly irradiate the flashlight to observe whether the pupil still has the function of contracting. Inthenextstep, participatewereinstructedtotakeobjective optometryandsubjective refraction. Wetakethe cycloplegic refraction inordertoprovidethe fully corrected soft contact lenses forparticipate to achievethe best corrected visual acuity during the measurement. Wearesorryabouttheunclearexpression. Actually, cycloplegic refraction isaconventionalexaminein Optometry clinic. Werecruitteenagers with cycloplegic refraction andensuretheywearappropriatelensduringourexperiment. Sothecycloplegic refraction isnot a major projectinourstudy. We -wouldliketodeletethissentencewithyourpermission. With the statementinline152-153 ispreciseenough.

We would like to enrich thiscontentinthemethodsandresultsinline149-151 andline195-196 respective.

Slit-lamp examination mainly included the examination of cornea (corneal fluorescein staining), lens, tear film (BUT ) andconjunctiva, all participates did not find any ocular surface damage, and BUT did not change significantly after the experiment.

lines 242-243: The sentence "Accommodative insufficiency, accommodative ex-242 cess and accommodative infacility were the main cause" needs a reference

Wearegratefulforthiscommentandaddthereference at the specified location.

lines 243-244: The sentence "Many patients alleviated the 243 symptoms and improved their vision function by accommodative training" needs a reference

 Wearegratefulforthiscommentandaddthereference at the specified location.

lines 246-247: "Our study revealed that the dynamic lighting can improve the subject’s accommodative function, which might be a new vision therapy for accommodation" In my opinion, this statement is not supported by the results of the study

 The reviewer’s statementiscorrect. Itistoo far to discuss theeffectofvisiontrainingbydynamiclighting. Thepresentstudyfirst demonstrate thefactthatdynamiclightingcanaltertheaccommodationofteenagers. Butthetrainingeffectneedstobefurtherresearch. Sowewouldliketodeletethissentenceasyoursuggestion and replacethefollowingsentence"Our study revealed that the dynamic lighting can alter the accommodationofteenagers, whichmayplayapotentialroleinaccommodationvisiontraininglikeflipper.”

And we would like to add this sentence in the end of this paragraph“However, the trainingeffect of dynamiclighting and the suitableparametersneedstobefurtherresearch”

lines 253-267: In my opinion, this part of the discussion could be omitted

 Wearegratefulforthesuggestionandhaveconsidereditcarefully.Itis unfavorable for readers tounderstandthearticle while this part exists. Wewouldliketodeletethisparagraphasyoursuggestion.

line 257: this hypothesis has not been previously cited

We thank the reviewer for this comment. As you say,thispresentationisless rigorousandinaccuracy. Wewouldliketodeletethissentenceasyoursuggestion.

  1. Wolffsohn JS, Gilmartin B, Mallen EA, Tsujimura S: Continuous recording of accommodation and pupil size using the Shin-Nippon SRW-5000 autorefractor. Ophthalmic & physiological optics : the journal of the British College of Ophthalmic Opticians (Optometrists) 2001, 21(2):108-113.
  2. Wolffsohn JS, O'Donnell C, Charman WN, Gilmartin B: Simultaneous continuous recording of accommodation and pupil size using the modified Shin-Nippon SRW-5000 autorefractor. Ophthalmic & physiological optics : the journal of the British College of Ophthalmic Opticians (Optometrists) 2004, 24(2):142-147.
  3. Day M, Gray LS, Seidel D, Strang NC: The relationship between object spatial profile and accommodation microfluctuations in emmetropes and myopes. J Vis 2009, 9(10):5 1-13.

Reviewer 2 Report

This manuscript describes an interesting idea of changing the pupil diameter using light intensity to adjust amplitude of accommodation.

However, there are some areas that are unclear. Some of the major areas are presented below; for more specific comments, please refer to the pdf file.

1. There needs to be more clarity on the topic (e.g. vision training has not been tested using this method so the title of the manuscript seems to be too far-reaching) and the participants being studied, e.g. study participants are 12-18 years olds (would 12 years olds be considered adults?) but some of the discussion is about presbyopia and myopia.

2. The rationale for the parameters of the experiment need to be discussed in more detail, e.g. why the dynamic lighting steps were chosen, why the range of lighting intensities chosen, how comparable is 1000 lux in the static lighting vs the 5-3000lux variable lighting etc.

3. How safety is measured for this study is not discussed in the methods and appears to not be fully encompassing, e.g. mainly looking at anterior eye health but does not consider symptoms that participants may experience (which many of the digital eye strain complaints are).

4. It is unclear how the change in amplitude of accommodation and pupil size in the two type of lighting conditions (static and dynamic) are calculated; some raw data for the different conditions may help with understanding. A difference of  1.01 +/- 0.53D was found for the dynamic group but it is unknown whether this amount of change is useful in a vision training sense (e.g. for vision training, +/-2.00D flippers or -3.00DS lenses may be used to train accommodation).

5. The link between pupil size and changing of depth of focus may also affect amplitude of accommodation, but this has not been discussed. There also appears to be some confusion over the use of accommodative response (is this talking about amplitude of accommodation like in the methods, or is it talking about the accuracy of accommodation like accommodative lag in the discussion?

6. Some references are missing and some do not match the information that is being referenced.

7. Grammar is not consistent throughout the manuscript. 

Author Response

This manuscript describes an interesting idea of changing the pupil diameter using light intensity to adjust amplitude of accommodation.

However, there are some areas that are unclear. Some of the major areas are presented below; for more specific comments, please refer to the pdf file.

Thank the reviewers for these precious comments concerning our manuscript entitled “Is accommodation influenced by dynamic ambient illumination and pupil size? A vision therapy for accommodation at home during COVID-19”. These comments are all valuable and very helpful for revising and improving our paper, as well as the important guiding significance to our research. We have studied comments carefully and have made corrections which we hope to meet with approval. Wewouldliketoreplyinthistextincludingresponseofthe PDF andsuggestioninthistext. we are also grateful fortheeditorof IJERPH, sincethereview was meticulous and helpful.

1.There needs to be more clarity on the topic (e.g. vision training has not been tested using this method so the title of the manuscript seems to be too far-reaching) and the participants being studied, e.g. study participants are 12-18 years olds (would 12 years olds be considered adults?) but some of the discussion is about presbyopia and myopia.

1.Wearegratefulforthiscomment. Vision training has not been tested using method inthisstudy. Itistoo far to discuss theeffectofvisiontrainingbydynamiclighting as you say. Sowewouldliketodeletethissentenceasyoursuggestion and replacethefollowingsentence(ex: line261-262 indiscussionandline83-84 inintroduction). Our study revealed that the dynamic lighting can improve the subject’s accommodative function, which might be a new vision therapy for accommodation"Our study revealed that the dynamic lighting can alter the accommodationofteenagers, whichmayplayapotentialroleinaccommodationvisiontraininglikeflipper.”And we would like to add this sentence in the end of this paragraph“However, the trainingeffect of dynamiclighting and the suitableparametersneedstobefurtherresearch”.

2.Andtheprevious title was less rigorousandinaccuracytosomeextent. Wewouldliketoalterourtitleto“Accommodation can be influenced by dynamic ambient illumination and pupil size in children” with your permission”.

3.weareverysorryaboutthemistake. wewouldliketochangethiswordto “teenagers”

  1. The rationale for the parameters of the experiment need to be discussed in more detail, e.g. why the dynamic lighting steps were chosen, why the range of lighting intensities chosen, how comparable is 1000 lux in the static lighting vs the 5-3000lux variable lighting etc.

We thank the reviewer for this comment. The range andstepwechoosefor dynamic ambient illumination baseonthe preliminary experiment. Although the illuminance of outdoor natural light can reach more than 20,000 lux people maynotfeeluncomfortable. Buttheindoor lighting is not uniform, somepatientin preliminary experiment complainglarewhilethe illuminance exceeds 3500-4000lux. Second,the pupilsizeand accommodation responseis 2.85±0.60mm and 2.50±0.88 respective. The accommodation responsein 40cmdistanceis 2.5D andthepupilsizeof 2.85mm isverysmall, increasingtherangeof illuminance beyond 5-3000lux may not achieve more variation inbothpupiland accommodation. Thestepof 5, 100, 200lux is thecommonuse illuminance inusuallife. When turning off the lights and closing the curtains, the dim light in room is nearly 5lux. When only indoor ceiling lamps are used for lighting, the illuminance of the tabletop is usually at 100-200lux. The illuminance of blackboard isrequiredtobemorethan 500lux, according to thehygienic standard for classroom lighting in primary and secondary schools in China (GB/T 7793-2010). The illuminance ofhallway measured at 150 cm fromclassroomdooris 1846lux, the illuminance ofshadebesideabuildingat 500 cm fromclassroomis 3140lux, andthe illuminance ofplaygroundis 2450lux. Wechoosethe 1000, 2000 and 3000lx becauseitrepresentsdifferent illuminance inourdailylifeandthe illuminance inindoor won't make peoplefeel uncomfortable[1]. Addition, as mentioned above, increasingtherangeof illuminance beyond 5-3000lux may not achieve more variation inbothpupiland accommodation. Of course, other steps can also be selected, such as 1500, 2500lux, because the most significant pupil and accommodation responses occur in the darkest or brightest illumination, so the difference is limited .

  1. How safety is measured for this study is not discussed in the methods and appears to not be fully encompassing, e.g. mainly looking at anterior eye health but does not consider symptoms that participants may experience (which many of the digital eye strain complaints are).
  • The reviewer’s statementiscorrect. The slit lamp examination should be mentioned in the methods section. We would like to enrich thiscontentinthemethodsandresultsinline149-151 andline195-196 respective.

Slit-lamp examination mainly included the examination of cornea (corneal fluorescein staining), lens, tear film ( BUT ) andconjunctiva.

All participates did not find any ocular surface damage, and BUT did not change significantly after the experiment.

  • Infact, theexamining physician interviewed the patients afterthedynamic ambient illumination forthesubjective perception including three parameters, clarity, comfort and glare. The value 0-9 represents the degree of feeling in a questionnaire. In addition, the research group also selected the most common reading desk lamps on the market as a comparison, named Desk Lamp B and Desk Lamp C, both of which are static light sources. Three kinds of illuminant were placed in the room to set up the test space. After the subjects entered and entered the room, they read under three illuminant in random order for 2 to 3 minutes. Finally, the subjects completed the "Table Lamp Comfort Questionnaire" we designed it ourselves.

The sharpness and comfort score of the dynamic ambient illumination is better than table lamp B and C, and the glare is similartothe table lamp B. Thesettingofthe table lamp B and C: 900lx and color temperature was 4000k for reading lamp B and those for reading lamp C were 500lx and 3000k

The resultshave shownthatparticipateundergo dynamic ambient illumination didnot manifest significant symptoms oranyuncomfortable complaint. Wedidnotpresentthispartofdatabecauseofthelimitation. Wedidnot adopt the international acknowledged visual discomfort scale like Conlon Visual Discomfort Survey or QOV.This may not be rigorous. Sowedecidenottoputforwardthispartofdata. Butin roughly speaking, wemaysaythatparticipateundergodynamic ambient illumination inpresentstudydidnotappearsignificantuncomfortablesymptoms. Wewouldliketoaddthissentence withyourpermission in line196-200 as follow:

“Theophthalmologist interviewed the patients afterthedynamic ambient illumination forthesubjective perception including three parameters, clarity, comfort and glare. Andfoundthatparticipateundergo dynamic ambient illumination didnot manifest significant symptoms oranyuncomfortable complaint.”

  1. It is unclear how the change in amplitude of accommodation and pupil size in the two type of lighting conditions (static and dynamic) are calculated; some raw data for the different conditions may help with understanding. A difference of 1.01 +/- 0.53D was found for the dynamic group but it is unknown whether this amount of change is useful in a vision training sense (e.g. for vision training, +/-2.00D flippers or -3.00DS lenses may be used to train accommodation).

The difference in the accommodative response measuredunder the maximum illumination and the minimum with the same visual stimulus, represented the amount of accommodative training. As shown inFig3 and Table 2, We found that the highest accommodative response isrelatedtothe brightest illumination, on the contrary,the lowest accommodative response isrelatedtothedarkestillumination. Sothe change in amplitude of accommodation and pupil size isthe difference value betweenthistwo illumination.

Thus, we designed a kind of artificial lighting which was dynamic controlled by microcomputer and its light intensity can dynamically change in the range of 0~3000lux automatically and smoothly. We wishto carry on training on the accommodation function via this dynamic ambient illumination, since study under the dynamic lighting need without initiativeandthe training can be completed unconsciously. The patient would have great compliance, especially in childrenandteenagers.

However, vision training has not been tested using method inthisstudy. Itistoo far to discuss theeffectofvisiontrainingbydynamiclighting. Thepresentstudy is a prior study, whichfirst demonstrate thefactthatdynamiclightingcanaltertheaccommodationofteenagers. Butthetrainingeffectneedstobefurtherresearch. Sowewouldliketodeletethissentenceasyoursuggestion and replacethefollowingsentence(ex: line261-262 indiscussionandline83-84 inintroduction). Our study revealed that the dynamic lighting can improve the subject’s accommodative function, which might be a new vision therapy for accommodation"Our study revealed that the dynamic lighting can alter the accommodationofteenagers, whichmayplayapotentialroleinaccommodationvisiontraininglikeflipper.”And we would like to add this sentence in the end of this paragraph“However, the trainingeffect of dynamiclighting and the suitableparametersneedstobefurtherresearch”.

  1. The link between pupil size and changing of depth of focus may also affect amplitude of accommodation, but this has not been discussed. There also appears to be some confusion over the use of accommodative response (is this talking about amplitude of accommodation like in the methods, or is it talking about the accuracy of accommodation like accommodative lag in the discussion?

Thank the reviewers for these precious comments. Themaintargetofaccommodationinpresentstudyis accommodative response andthe amplitude of accommodation under the maximum illumination and the minimum illumination (difference value ofthe accommodative response), we talked aboutthe accommodative lag indiscussioninordertomanifestthedegenerationofaccommodation. Sincewedidnot execute theexperimentinadultexceed 40. Thispartofextentcanberemovedifyousuggestus. 

This research is a self-controlled study. Wedidnotmeasurethe amplitude of accommodation, wejustmeasurethe accommodation underdifferentillumination.

  1. Some references are missing and some do not match the information that is being referenced.

Wearegratefulforthiscommentandaddthereference at the specified location.

  1. Grammar is not consistent throughout the manuscript

Thank the reviewers for these precious comments concerning our manuscript. Your review was meticulous and helpful. We will improve the grammar and writing skills of any sentences in the text that need to be revised. We are also always ready to amend it as necessary.

1.What power of flippers were used for the training?

Theyusedthe accommodative flippers at±2 D

2.Why just the right eye?

 Because parameters of both eyes, including accommodation and pupillary responses are similar and synchronized, only one eye was selected for measurement

 3.Any measures of pupil size or reactions? Disorders of the pupils?

We will obtain pupil-related data when performing objective refraction before enrollment, and patients with abnormal pupils are also excluded from the group.

4.What was the periodic time used for this study?

Wearesorryabouttheunclearstatement. This is a cross-sectional study. we describe it inline101-102.

  1. Was the patient a similar distance from both lighting systems?

All participates were at the same distance under different lighting conditions.

6.How is the UBM used to measure pupil diameter?

Weare very gratefulforthis good comment. Weusethe SRW-5001K and an ultrasound biological microscope (UBM) toexamine the accommodative response and pupil diameter. Infact, SRW-5001K can measure accommodative response and pupil sizeinthesametime. Continuouson-line analysis of the ring provides high (up to 60 Hz) temporal resolution of the refractive state toan accuracy of ,0.001 D. Pupil size can also be analysed to a resolution of ,0.001 mm.

We want to use UBM to visualize the alteredimage of pupil and ciliary muscles under different illumination, so the previous expression may be inaccurate. The measurement of pupil diameter as same as the accommodation response were through SRW-5001K. UBM shows the pupil and accommodation status in different illumination. The pupil data measured by UBM is almost the same as that of SRW-5001K, so it is enough to keep one of them. Wewouldliketorevisethesentenceinline141 andline134/137/145 asshownbelow:

line141: Then the pupil and ciliary muscles diameter was measured image was captured under the same conditions by UBM (Model SW-3200L).

line134:Accommodative response and pupil diameter was measured by….

line137:their accommodative response and pupil diameter were measured seven times….

line145: the pupil diameter anterior segment image of the right eye was measured.

Andwewanttoaddsentenceasfollowinline120-122.

Previous studies have confirmed the validity of the instrument measuring accommodation. The power refractor was calibrated priorto the study.

7.I am slightly confused about what is the minimum and maximum amplitude of accommodation/pupil diameter; e.g. for static 1000lux, is it out of the 7 measurements, the largest measurement minus the smallest measurement? Then there would be no standard error as there would only be one number. For Dynamic, it would be assumed that at 5 lux would be the largest pupil diameter, while at 3000lux, the pupil would be smallest; so if you are taking just the maximum and minimum across the 7 different light intensities, then the comparison would be different, e.g. 1000lux vs 0-3000lux

Asshownin Table 2, Under the same light intensity, subjects' accommodation and pupillary responses fluctuated within a certain range. A phenomenon termed accommodative microfluctuations, the accommodationof the eye is not stable but constantly varies at a rapid speed within a range of approximately 0.50 D, soas in pupilsize.

8.How about mean pupil diameter?

The mean pupil diameter wasshown in Table 2 underdifferent light intensity.

9.Would the difference be due to the difference in maximum light level? e.g. dynamic group can be up to 3000 lux, while the static group is 1000 lux only.

Yes, the difference be due to the difference in maximum and minimal light level.

10.This does not seem relevant because this study was looking at participants who are 12-18 years of age. It would be expected that in presbyopes, there are also changes in pupil diameter, which would change the depth of focus.

we talked aboutthe accommodative lag indiscussioninordertomanifestthedegenerationofaccommodation. Sincewedidnot execute theexperimentinadultexceed 40. Thispartofextentcanberemovedifyousuggestus.

11.Could taking more breaks from near work help?

Yes, it helps.

12.Will a patient be annoyed or tired from doing tasks in a room where the lighting goes dim and bright?

As mentioned above, this phenomenon did not occur in our experiments.

13.This study was looking at increasing the difference in minimum to maximum pupil size but did not talk about using optics or drugs to change the size of the pupils.

It is unclear what the focus of this paragraph is; it starts off with presbyopia, but then talks about pupil diameters and lag of accommodation in myopia.

   Wearegratefulforthesuggestionandhaveconsidereditcarefully.Itis unfavorable for readers tounderstandthearticle while this part exists. Wewouldliketodeletethisparagraphasyoursuggestion.

  1. Wu PC, Chen CT, Lin KK, Sun CC, Kuo CN, Huang HM, Poon YC, Yang ML, Chen CY, Huang JC et al: Myopia Prevention and Outdoor Light Intensity in a School-Based Cluster Randomized Trial. Ophthalmology 2018, 125(8):1239-1250.

Reviewer 3 Report

This is very nice study by Hanyang Yu et al. about a  potential vision therapy for accommodation at home. The basic idea is inetersting, however, it is not clear to this reviwer why UBM and not other non-contact imaging of the anterior segment was performed. As good as the idea is, the authors should more emophasize that a larger amount of oatients is needed to draw final conclusions from this study.

Author Response

This is very nice study by Hanyang Yu et al. about a potential vision therapy for accommodation at home. The basic idea is inetersting, however, it is not clear to this reviwer why UBM and not other non-contact imaging of the anterior segment was performed. As good as the idea is, the authors should more emophasize that a larger amount of oatients is needed to draw final conclusions from this study.

Thank the reviewers for these precious comments concerning our manuscript entitled “Is accommodation influenced by dynamic ambient illumination and pupil size? A vision therapy for accommodation at home during COVID-19”. These comments are all valuable and very helpful for revising and improving our paper, as well as the important guiding significance to our research. We have studied comments carefully and have made corrections which we hope to meet with approval.

Weare very gratefulforthis good comment. Weusethe SRW-5001K and an ultrasound biological microscope (UBM) toexamine the accommodative response and pupil diameter. Infact, SRW-5001K can measure accommodative response and pupil sizeinthesametime. Continuouson-line analysis of the ring provides high (up to 60 Hz) temporal resolution of the refractive state toan accuracy of ,0.001 D. Pupil size can also be analysed to a resolution of ,0.001 mm.

We want to use UBM to visualize the alteredimage of pupil and ciliary muscles under different illumination, so the previous expression may be inaccurate. The measurement of pupil diameter as same as the accommodation response were through SRW-5001K. UBM shows the pupil and accommodation status in different illumination. The pupil data measured by UBM is almost the same as that of SRW-5001K, so it is enough to keep one of them. Wewouldliketorevisethesentenceinline141 andline134/137/145 asshownbelow:

line141: Then the pupil and ciliary muscles diameter was measuredimage was captured under the same conditions by UBM (Model SW-3200L).

line134:Accommodative response and pupil diameter was measured by….

line137:their accommodative response and pupil diameter were measured seven times….

line145: the pupil diameter anterior segment image of the right eye was measured.

Andwewanttoaddsentenceasfollowinline120-122.

Previous studies have confirmed the validity of the instrument measuring accommodation. The power refractor was calibrated priorto the study.

This manuscript is a resubmission of an earlier submission. The following is a list of the peer review reports and author responses from that submission.